# Emergent Recurrent Extension Phase Transition in a Quasiperiodic Chain

Shan-Zhong Li[1,2] and Zhi Li[1,2]

[1] Key Laboratory of Atomic and Subatomic Structure and Quantum Control (Ministry of Education), Guangdong Basic Research Center of Excellence for Structure and Fundamental Interactions of Matter, School of Physics, South China Normal University, Guangzhou 510006, China
[2] Guangdong Provincial Key Laboratory of Quantum Engineering and Quantum Materials, Guangdong-Hong Kong Joint Laboratory of Quantum Matter, Frontier Research Institute for Physics, South China Normal University, Guangzhou 510006, China

## Abstract

We study $p$-wave superconducting quasiperiodic chains with staggered potential. The result shows a counter-intuitive phase transition phenomenon, i.e., recurrent extension phase transition (REPT). By analyzing the participation ration and scaling behavior, we prove the existence of REPT phenomenon, which, in concrete terms, means that the system will repeatedly return from the intermediate phase to the extended phase as the quasiperiodic or staggered strength grows. Furthermore, our finding is also quite different from the traditional understanding of intermediate phase (composed only of the pure extended phase and pure localized phase) in that, the new intermediate phase described here, stemming from the competition between staggered potential and $p$-wave pairing, actually falls into three types by bringing in the critical phase. To be specific, the new intermediate phases are composed of the critical + extended states, the critical + localized states, and the critical + extended + localized states, respectively.

# 1   Introduction

In the late 1950s, G. Feher and E. A. Gere of Bell Laboratories first discovered the relaxation of electron spin [1, 2]. To explain this phenomenon, P. W. Anderson proposed the famous Anderson Localization theory that when metal doping exceeds the threshold value, conductivity of the system will change dramatically from the metallic phase to the insulating phase [3]. Later in the 1960s, N. F. Mott pointed out that the localized state and the extended state can coexist in some cases, giving rise to mobility edge in the system [4]. According to the scaling theory of disordered systems, when the system dimension $D < 3$, any strength of disorder will nudge the system into the localized phase, leaving the system's metallic-insulating phase transition to be destroyed [5–8]. However, when $D = 3$, such phase transition is allowed by increasing the disorder strength and adjusting the Fermi energy before the critical disorder strength due to the presence of the mobility edge. Disorder-induced Anderson localization in low-dimensional systems is trivial, but its application to the study of topological phase transitions [9–17], many-body localization [18–24], and etc. reveals many novel phenomena.

Compared with random disordered systems, quasiperiodic systems that entail less numerical computation and relatively convenient analytical deduction have been widely used to study Anderson localization and mobility edges. Meanwhile, quasiperiodic systems have been implemented on many experimental platforms, including photonic crystal [25–29], optical waveguide arrays [30–32], cold atom experiments [33–38], and other related fields [39–41]. As a typical low-dimensional quasiperiodic model, the Aubry-André-Harper (AAH) chain has been extensively studied over the past few decades [42, 43]. This can be attributed to the self-duality property of the AAH model, which means the distribution of its eigenfunctions in both real space and momentum space is exactly the same for the critical point. Based on this, one can easily obtain the critical point of phase transition through analytical deduction [43], so as to well grasp the characteristics of the extended and localized phase transition.

Previous theoretical studies have suggested that the mobility edge of AAH model can be achieved by introducing a long-range hopping [44–46], a dimer hopping [47, 48], a spin-orbit coupling [49, 50], or a controlled quasiperiodic potential [51–55]. Then in recent experiments, the above mobility edge phenomenon has been realized in different platforms one after another [37, 56, 57]. Besides, duality transformation [44, 51, 58–60] and the famous Avila global theory [54, 61–73] provide us with an analytical alternative to deal with mobility edges. In addition to the mobility edge caused by the coexistence of the traditional extended and localized state, quasiperiodic systems are capable of inducing novel mobility edges. As mentioned above, this is because the phase transition critical point in AAH model features self-duality, which causes the corresponding eigenstate to be a multifractal critical state, neither extended nor localized. Here, the system shows a multifractal critical phase, obviously different from the energy level statistics [74, 75], wave function distribution [76, 77], and dynamic behavior of the pure extended and localized phases [78, 79]. Though the multifractal critical behaviors of the system at the critical point seem very attractive, its application in experiments is limited due to the highly demanding techniques in preparation. Recent years have witnessed lively discussions on how to stretch the critical point out to a critical region, with an aim to improve the robustness of the system in the multifractal critical phase. By introducing $p$-wave superconducting pairing [80–83], spin-orbit coupling [84, 85], off-diagonal quasiperiodic hopping [61, 63, 86–92] and other means [62, 93, 94], the multifractal critical region has been

successfully wrought. The emergence of critical states has enriched the concept of mobility edges, and novel energy-dependent mobility edges for the coexistence of critical, extended and localized states have been predicted in many studies in recent years [84]. In addition to localization phase transitions, the AAH model is also valuable for the study of topological phases in quasicrystals, for it can be mapped to the two-dimensional integer quantum Hall effect by means of a continuous U(1) metric transformation [87–92, 95–105].

So far, fruitful results have been achieved in the study of quasiperiodic systems, and the most remarkable among them is multiple re-localization, i.e., by manipulating quasiperiodic parameters, repeated re-localization can emerge in the system [47, 48, 55, 85, 106–113]. However, no paper has yet proved or disproved whether an ever-growing quasiperiodic strength will bring the system back from the localized or intermediate phases (coexistence of different states) to the extended phase. So what is the real-world situation? To find out the answer, we study the phase transition of the $p$-wave superconducting paired AAH model with staggered on-site potential. The results demonstrate the novel intermediate-extended phase transition, which proves the system will indeed revert from the intermediate phase to the extended phase as the quasiperiodic strength grows. Furthermore, the introduction of staggered potentials also enables the emergence of new-types mobility edges.

The paper is organized as follows. We introduced the model in Sec. 2. We discuss observable quantities and the recurrent extension phenomenon in Sec. 3. Then, we investigate the emergent intermediate phases through various observable quantities and the corresponding scaling analysis in Sec. 4. Main findings of this paper are concluded in Sec. 5.

## 2  Model

We start from the $p$-wave superconducting paired AAH model with the staggered on-site potential and the corresponding Hamiltonian reads

$$H = \sum_{j=1}^{N-1}(Jc_j^\dagger c_{j+1} + \Delta c_j^\dagger c_{j+1}^\dagger + \text{H.c.}) + \sum_{j=1}^{N}(V_j + W_j)c_j^\dagger c_j, \tag{1}$$

where $c_j$ ($c_j^\dagger$) is the annihilation (generation) operator at site $j$, $N$ is the total number of lattices, $J$ corresponds to the strength of the nearest neighboring hopping, and $\Delta$ denotes the intensity of $p$-wave pairing. The on-site potential is composed of two parts. One is the quasiperiodic part $V_j = \lambda\cos(2\pi\alpha j + \theta)$, where $\lambda$ stands for the quasiperiodic strength, $\alpha$ and $\theta$ indicate the irrational number and phase shift [114], respectively. The other one is the staggered potential $W_j = \eta(-1)^j$, where $\eta$ refers to the intensity of the staggered potential.

Under the condition of $\Delta$, $\eta = 0$ and $\lambda > 0$, Eq. (1) can be reduced to the standard AAH model [42, 43], whose critical point of phase transition is $\lambda_c = 2J$. When $\eta = 0$ and $\Delta$, $\lambda > 0$, the system becomes the $p$-wave superconducting paired AAH model and the phase diagram is exhibited in Fig. 1, where the line of $\lambda = 2|J + \Delta|$ separates the localized from the critical phase, while the line of $\lambda = 2|J - \Delta|$ draws a distinction between the critical and the extended phases [80]. When $\eta$, $\lambda > 0$ and $\Delta = 0$, however, the system will exhibit re-entrant localized phase transition [55].

In the particle-hole picture, one can diagonalize the Hamiltonian Eq. (1) by Bogoliubov-de Gennes (BDG) transformation [115], and then we obtian

$$H = \sum_{n=1}^{N}\varepsilon_n(\gamma_n^\dagger\gamma_n - \frac{1}{2}), \tag{2}$$

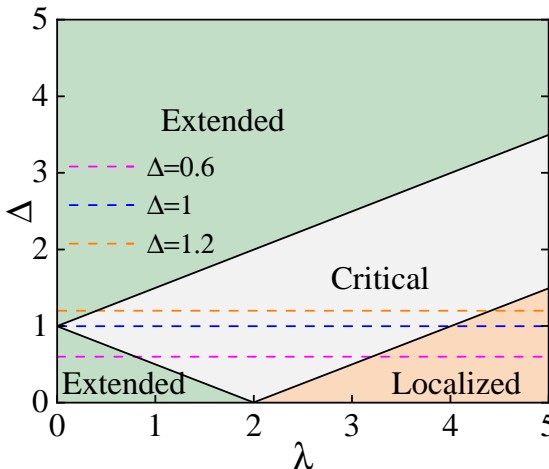

Figure 1: (color online). The phase diagram of the standard $p$-wave paired AAH model ($\eta = 0$ in Eq. 1). The green, grey, and wheat regions represent the extended, critical, and localized phases, respectively. Since we mainly discuss the effect of staggered potential, hereafter we focus on the $\eta - \lambda$ phase diagram with fixed $\Delta = 0.6$ (the purple dashed line), $1$ (the blue dashed line) and $1.2$ (the orange dashed line), respectively.

where $\gamma_n = \sum_{n=1}^{N}(u_{n,j}^{*}c_j + v_{n,j}c_j^{\dagger})$ with the energy level index $n = 1, 2, \ldots, N$. $u_{n,j}$ and $v_{n,j}$ denote the two components of the wave function at site $j$. The eigenspectra $\varepsilon_n$ and the corresponding eigenstates $|\psi_n\rangle = (u_{n,1}, v_{n,1}, u_{n,2}, v_{n,2}, \ldots, u_{n,N}, v_{n,N})^T$ are determined by Schrödinger equation $H|\psi_n\rangle = \varepsilon_n|\psi_n\rangle$, where Hamiltonian $H_n$ is a matrix of $2N * 2N$. The expression takes the form

$$H = \begin{pmatrix} A_1 & B & 0 & 0 & 0 & \ldots & C \\ B^{\dagger} & A_2 & B & 0 & 0 & \ldots & 0 \\ 0 & B^{\dagger} & A_3 & B & 0 & \ldots & 0 \\ \vdots & \ddots & \ddots & \ddots & \ddots & \ddots & \vdots \\ 0 & \ldots & 0 & B^{\dagger} & A_{N-2} & B & 0 \\ 0 & \ldots & \ldots & 0 & B^{\dagger} & A_{N-1} & B \\ C^{\dagger} & \ldots & \ldots & \ldots & 0 & B^{\dagger} & A_N \end{pmatrix}, \tag{3}$$

where

$$A_j = \begin{pmatrix} V_j + \eta(-1)^j & 0 \\ 0 & -V_j - \eta(-1)^j \end{pmatrix}, \tag{4}$$

$$B = \begin{pmatrix} -t & -\Delta \\ \Delta & t \end{pmatrix}. \tag{5}$$

For periodic boundary conditions (PBCs),

$$C = \begin{pmatrix} -t & \Delta \\ -\Delta & t \end{pmatrix}, \tag{6}$$

while for open boundary conditions (OBCs),

$$C = 0. \tag{7}$$

The dimension of the corresponding BDG Hamiltonian is $2N$. During numerical calculation, we set the system size $N = F_m$, where $F_m$ stands for the $m$-th Fibonacci number,

which satisfies $F_{m+1} = F_m + F_{m-1}$, and $F_0 = F_1 = 1$. Besides, the irrational number is set as $\alpha = F_{m-1}/F_m$. Without loss of generality, we take $J = 1$ as the unit of energy and select periodic boundary conditions in the process of calculation. Since the value of $\theta$ has no qualitative impact on the system, we set $\theta = 0$ in the following analysis.

## 3 Recurrent Extension Phase Transition

### 3.1 Phase diagram for $\Delta = 0.6$

Inverse participation ratio (IPR) and normalized participation ratio (NPR) are the core observables to determine the localization properties of the system [47, 52, 55, 116], whose definitions take the form

$$\xi_n = \frac{\sum_j (u_{n,j}^4 + v_{n,j}^4)}{\sum_j (u_{n,j}^2 + v_{n,j}^2)}, \ \zeta_n = \left[ N \frac{\sum_j (u_{n,j}^4 + v_{n,j}^4)}{\sum_j (u_{n,j}^2 + v_{n,j}^2)} \right]^{-1}, \tag{8}$$

where $n$ represents the eigenstate index, $u_{n,j}$ and $v_{n,j}$ denote the expansion coefficients of the $n$-th eigenstate at site $j$ on the BDG basis. By calculating the average IPR $\overline{\xi} = \frac{1}{N} \sum_n \xi_n$ and the average NPR $\overline{\zeta} = \frac{1}{N} \sum_n \zeta_n$, one can determine the extended, localized and intermediate phases, respectively. In concrete terms, the extended (localized) phase corresponds to $\overline{\xi} \sim 0$ ($> 0$) and $\overline{\zeta} > 0$ ($\sim 0$). For the intermediate phases, however, both $\overline{\xi}$ and $\overline{\zeta}$ are of finite values due to the coexistence of different states in the system [52]. Based on this, one can define

$$\kappa = \log_{10}(\overline{\xi} \times \overline{\zeta}) \tag{9}$$

to distinguish the pure phase (pure extended or pure localized phase) from the intermediate phase [47, 55, 116]. In the following analysis, we set $N = 610$, where the dimension of the BDG Hamiltonian matrix corresponding to the system is greater than $10^3$. So the intermediate phase (pure phase) corresponds to $\kappa \to 0$ ($\kappa \to -3$).

The $\lambda - \eta$ phase diagram for $\Delta = 0.6$ is plotted in Fig. 2. As is shown, $\eta = 0$ corresponds to the case with no staggered potential, i.e., the system will gradually move from the extended, critical, and eventually into the localized phase, with the critical points of phase transition being $0.8$ and $3.2$ (black dashed lines), respectively, which corresponds to purple dashed line in Fig. 1.

The introduction of staggered potential will bring about richer phases. Firstly, different from the pure extended phase that emerges in the standard $p$-wave superconducting AAH model, the introduction of staggered potential will give rise to mobility edges and the intermediate phase for $\lambda < 0.8$. Specifically, within a certain parameter range (around $\lambda \sim 0.7$), the emergent recurrent extension occurs, i.e., the system transforms from the extended phase to intermediate phase and then back to the extended phase with an increasing quasiperiodic strength $\lambda$ (red dashed line in Fig. 2). This REPT phenomenon can be clearly seen from the $\kappa$ and average IPR $\overline{\xi}$ phase diagrams on the $\lambda - \eta$ plane, as shown in Fig. 2. In the figure, one can also find that with the increase of $\eta$, there are still several other regions in the system where REPT occurs, however, the value of $\eta$ should not be taken too large. We show in the following part of the paper that when $\eta$ tends to infinity, REPT phenomenon will disappear and the system will display a clear transition boundary between extended and localized phase.

Secondly, in the region of $0.8 < \lambda < 3.2$ (between two black dashed lines, when $\eta = 0$ is of critical phase), when the staggered potential intensity $\eta$ is relatively large ($\eta > 3$), the system is easier to enter the localized phase with the increasing $\eta$. However, when $\eta$ is relatively small, there exists a region with a large fluctuation of $\kappa$, which is marked by red

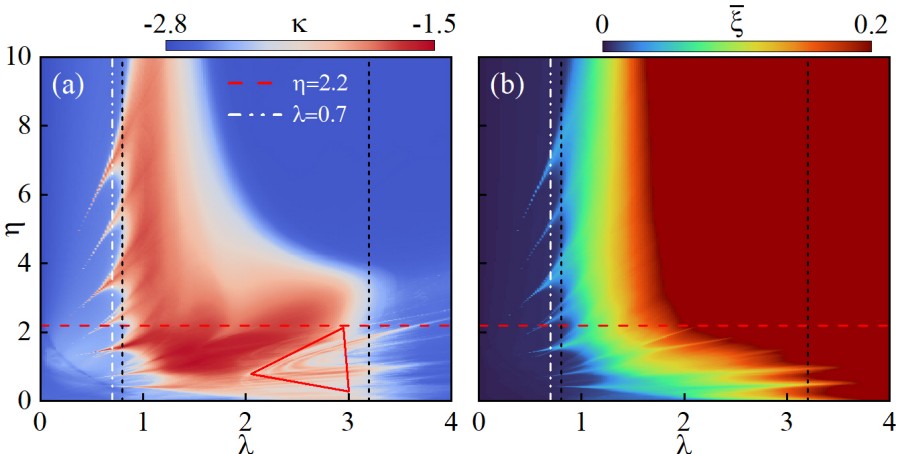

Figure 2: (color online). The (a) $\kappa$ and (b) $\overline{\xi}$ phase diagram in $\lambda - \eta$ plane with $\Delta = 0.6$. The red and white dashed lines correspond to slices with $\eta = 2.2$ and $\lambda = 0.7$, respectively. The two black dashed lines mark the critical points of phase transition from the extended to critical and from critical to localized phases of standard $p$-wave paired AAH model ($\eta = 0$), respectively. Throughout, we set the system size $N = 610$.

triangular in Fig. 2(a). Due to the unusual features of fluctuation, one can naturally expect new intermediate phases and the corresponding mobility edges in this region. We will discuss such regions in the next section.

Thirdly, for $\lambda > 3.2$, the quasiperiodic potential will prevail. Now that the staggered on-site potential does not have much leverage on it, the system always stays in the localized phase.

Besides, the fractal dimension serves as a valid indicator to identify the mobility edge and distinguish different phases. The corresponding fractal dimension of $n$-th eigenstate reads

$$\Gamma_n = -\frac{1}{2}(\frac{\ln \sum_j u_{n,j}^4}{\ln 2N} + \frac{\ln \sum_j v_{n,j}^4}{\ln 2N}). \tag{10}$$

The extended, localized and critical phases correspond to $\Gamma_n \to 1$, $\Gamma_n \to 0$ and $0 < \Gamma_n < 1$, respectively [54, 84].

Moreover, the scaling index for multifractal analysis has a similar effect [80, 117–119]. The probability of the nth eigenstate on the site $j$ is represented by the wave function $\mathbb{P}_{n,j} = u_{n,j}^2 + v_{n,j}^2$, which satisfies the normalization condition $\sum_j \mathbb{P}_{n,j} = 1$. The scaling index of multifractal analysis $\beta_j$ for the $n$th eigenstate is defined by the probability measure $\mathbb{P}_{n,j}$ as

$$\mathbb{P}_{n,j} = (2N)^{-\beta_j^n}. \tag{11}$$

Since the occupation probability on all sites is $\mathbb{P}_j^n = 1/2N$ for a completely extended wave function, the corresponding scaling index $\beta_j^n = 1$. For a localized wave function, the occupation probability is non-zero at just a few sites, therefore $\beta^n \to 0$ for such occupied sites and $\beta^n \to \infty$ for the other sites. For a multifractal wave function, the scaling index $\beta^n$ is distributed in a finite interval $[\beta_{min}^n, \beta_{max}^n]$. Thus, by considering the thermodynamic limit $2N \to \infty$, one can characterize the localization properties of a wave function by $\beta_{min}^n$. To be specific, for $2N \to \infty$, $\beta_{min}^n = 1$ (0) indicates the extended (localized) states, whereas $0 < \beta_{min}^n < 1$ corresponds to the multifractal state.

In the next subsection, we will prove the REPT phenomenon through the results of $\kappa$, $\xi$, $\zeta$, $\Gamma$, and $\beta_{min}$. To better reveal the REPT, we consider two most representative cases in the

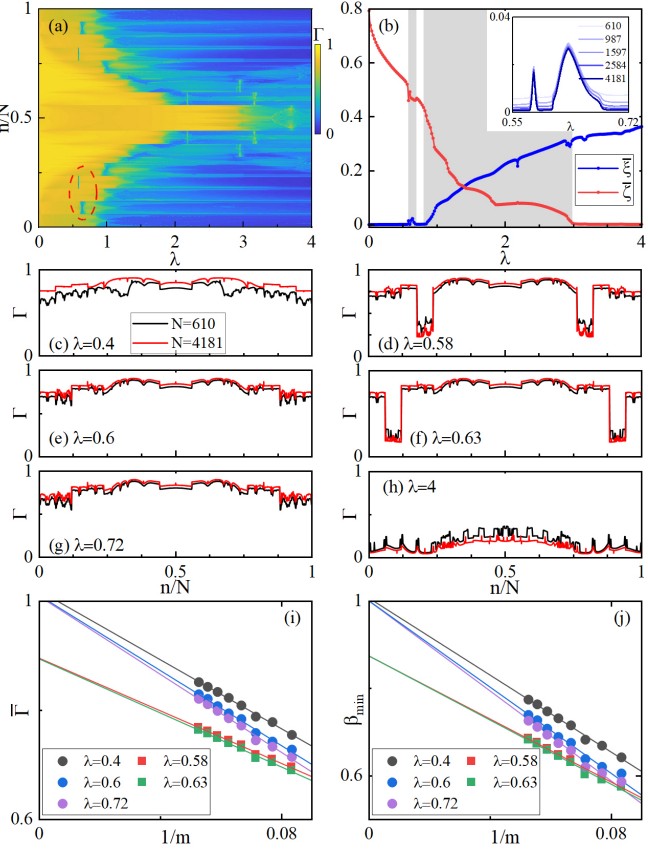

Figure 3: (color online). (a) Fractal dimension $\Gamma$ of all eigenstates as a function of $\lambda$ with $N = 610$. (b) $\overline{\xi}$ (blue) and $\overline{\zeta}$ (red) as a function of $\lambda$ with $N = 2584$. The inset show $\overline{\xi}$ for $N = 610, 987, 1597, 2584, 4181$, respectively. The behavior of $\Gamma$ for (c) $\lambda = 0.4$, (d) $\lambda = 0.58$, (e) $\lambda = 0.6$, (f) $\lambda = 0.63$, and (g) $\lambda = 0.72$ with respect to the system size. The scaling properties of (d) $\overline{\Gamma}$ and (e) $\beta_{min}$ as a function of $1/m$ for all eigenstates are provided, where the dashed line is the result of the linear fit and $m$ is the $m$th Fibonacci number, i.e. $N = F_m$. the Throughout, we set $\Delta = 0.6$ and $\eta = 2.2$.

above phase diagram: (1) Fix the staggered intensity at $\eta = 2.2$ (red dashed line) to study the effect of quasiperiodic potential on the system phase transition. (2) Fix the quasiperiodic potential at $\lambda = 0.7$ (white dashed line) to observe how the staggered potential will affect the phase transition.

## 3.2 $\lambda$-induced REPT

Now let's focus on the first case $\eta = 2.2$. We first calculate fractal dimension $\Gamma$ corresponding to different energy levels of the system, which is a good indicator in distinguishing different phases. One can notice that the situation of $\Gamma \to 0$ appears sporadically in the system near $\lambda = 0.6$ [shown by red circle in Fig. 3(a)], which means the system is not in the pure extended state and the intermediate phase is about to emerge [see Fig. 3(a)]. When $\lambda$ further increases, however, the system will resume its extension property. This is consistent with the phase diagram we have scanned through $\kappa$, i.e., emergent REPT will occur in the system. To better identify the emergent REPT, we plot the average IPR $\overline{\xi}$ and average NPR $\overline{\zeta}$ in Fig. 3(b). The $\overline{\xi}$ rises and falls multiple times around $\lambda = 0.6$, which consistently demonstrates that the

extended phase (white region) and the intermediate phase (gray region) will alternate in the region of quasiperiodic strength around $\lambda = 0.6$. We show in the inset the variation of the $\overline{\xi}$ with increasing system size near $\lambda = 0.6$. It can be seen that the $\overline{\xi}$ of the extended phase is constantly decaying towards 0, while the intermediate phase will gradually converge to form two distinct peaks. This is a sufficient explanation for the occurrence of REPT near $\lambda = 0.6$.

Besides, we show the scaling behavior of the fractal dimension $\Gamma$ of all eigenstates under different quasiperiodic intensities $\lambda$. As shown in the figure, as the size of the system increases, $\Gamma$ corresponding to eigenstates of the extended phase ($\lambda = 0.4, 0.6, 0.72$), the intermediate phase ($\lambda = 0.58, 0.63$) and the localized phase ($\lambda = 4$) tend to 1; partly to 1 and partly to 0; and to 0, respectively.

Behavior at finite sizes can initially determine the phase of the system. Further, we fit the case up to the thermodynamic limit by scaling analysis. We define the average fractal dimension $\overline{\Gamma} = \frac{1}{N} \sum_{n=1}^{N} \Gamma_n$ and average sacling index $\beta_{min} = \frac{1}{N} \sum_{n=1}^{N} \beta_{min}^{n}$ for all eigenstates. Scaling properties of different $\lambda$ are shown in Fig. 3(i)(j), where the result of the thermodynamic limit ($1/m \to 0$) is obtained by the linear fit and extrapolation method. For the extended phase (dots), both $\overline{\Gamma}$ and $\beta_{min}$ can reach 1 in the thermodynamic limit, while for the intermediate phase (squares), the corresponding $\overline{\Gamma}$ and $\beta_{min}$ can never reach 1 in the thermodynamic limit due to the coexistence of the localized phase and the extended phase.

The above results show that as $\lambda$ increases, the system will switch back to the extended state with the ever-increasing quasiperiodic strength, i.e., REPT occurs. This phenomenon of reentry into the extended state with the increase of quasiperiodic strength in weak quasiperiodic region is what we dub as the Type-I REPT phenomenon. From the phase diagram Fig. 2, one can see that the Type-I REPT phenomenon is relatively weak and the region where it first enters the intermediate phase is small.

### 3.3 $\eta$-induced REPT

Then we discuss the second case of REPT. By fixing the quasiperiodic strength $\lambda = 0.7$, one can see clearly how the REPT can be manipulated by tuning the staggered potential. Fig. 4(a) shows $\Gamma$ of all eigenstates versus staggered potential. The result reveals that the system repeatedly exhibits the coexistence of localization and extension properties with the increasing staggered potential, which confirms the multiple REPT. Further, we compute the average IPR $\overline{\xi}$ and NPR $\overline{\zeta}$, which both have similar periodic variations [see Fig. 4(b)]. In addition, we show the $\kappa$ for different system sizes as a function of $\eta$ in Fig. 4(c). For the extended phase, $\kappa \sim \log_{10}(1/2N)$ for large sizes, hence $\kappa$ decreases with increasing system size. For the intermediate phase, $\kappa \sim \log_{10}[O(1)]$, which remains constant as the system size increases. The occurrence of REPT is well illustrated in Fig. 4(b)(c). These indicator quantities consistently prove that the system can repeatedly switch between the extended phase and the intermediate phase. Finally, For the extended phase (dots), both $\overline{\Gamma}$ and $\beta_{min}$ can reach 1 in the thermodynamic limit, whereas for the intermediate phase (squares), it is between 0 and 1. We dub this multiple REPT phenomenon occurring with the changing staggered potential in the weak quasiperiodic region as Type-II REPT phenomenon.

### 3.4 The limits of the large $\eta$

Another interesting phenomenon occurs when $\eta$ is very large. From Fig. 2 we can see that as $\eta$ increases, the region of the extended and localized phases gradually expands and encroaches upon the region of the intermediate phase, a phenomenon that we also observe in other $\Delta$ parameters [see Fig. 6(a) and 8(a)]. It is as if to show that the system at $\eta = \infty$ has an exact extended-localized phase transition point.

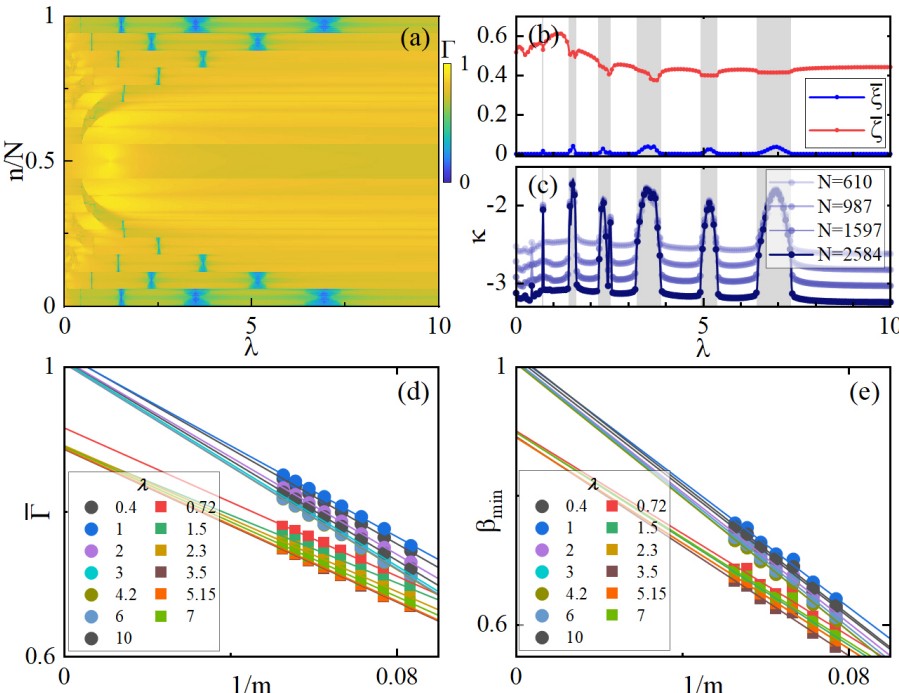

Figure 4: (color online). (a) Fractal dimension $\Gamma$ of all eigenstates as a function of $\lambda$ with $N = 610$. (b) $\overline{\xi}$ (blue) and $\overline{\zeta}$ (red) as a function of $\lambda$ with $N = 2584$. (c) $\kappa$ as a function of $\lambda$ for different system size $N$. The behavior of (d) $\overline{\Gamma}$ and (e) $\beta_{min}$ as a function of $1/m$ for different $\lambda$ with respect to the system size. where the dashed line is the result of the linear fit and $m$ is the $m$th Fibonacci number. Throughout, we set $\Delta = 0.6$ and $\lambda = 0.7$.

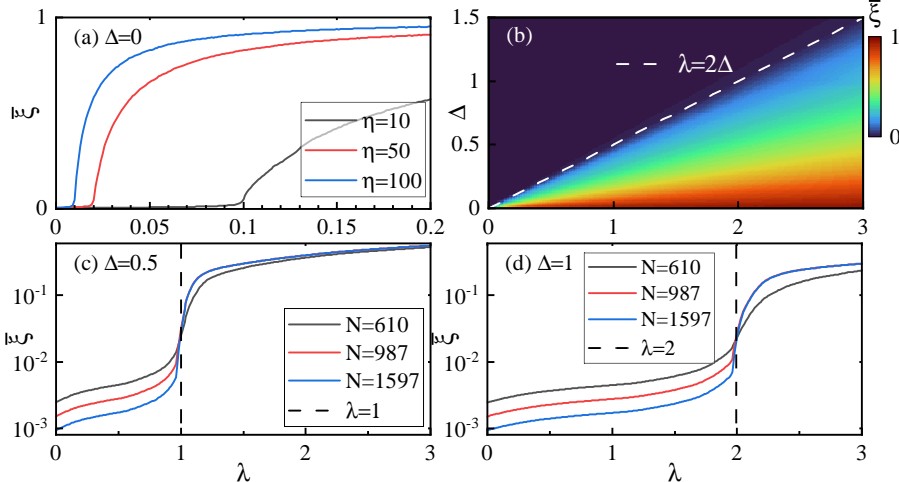

Figure 5: (color online). (a) The average IPR $\overline{\xi}$ of all eigenstates as a function of $\lambda$ for different $\eta$ with system size $N = 610$ and $\Delta = 0$. (b) The average IPR $\overline{\xi}$ in the $\lambda - \Delta$ plane for $\eta = 100$ and $N = 610$, where white dashed line is $\lambda = 2\Delta$. The average IPR $\overline{\xi}$ as a function of $\lambda$ for (c) $\Delta = 0.5$ and (d) $\Delta = 1$ with different system sizes.

First, we considered the case of $\Delta = 0$, and Eq. 1 returns to the AAH model with the staggered potential. In this case, the system will exhibit multiple reentrant localization phenomenon. Besides, one can find that staggered potential will enhance AAH model's localization properties in the extended region ($\lambda < 2$), making it easier for the system to enter the localized phase. Especially when $\eta$ is large, even very small quasiperiodic potential can make the system localized [55]. As shown in Fig. 5(a), we demonstrate the average IPR $\overline{\xi}$ as a function of $\lambda$ for $\Delta = 0$ at different $\eta$. In the case of a large $\eta$, the localization phase transition point exhibits $\lambda = 1/\eta$. By analogy, when $\eta = \infty$, an arbitrarily small $\lambda$ can induce a localization phase transition.

However, with the introduction of $p$-wave superconductivity, as shown in Fig. 5(b), the localization phase transition point at $\eta = 100$ exhibits the same behavior as when $\lambda = 2\Delta$ (where $1/\eta = 0.01$ is is small enough to be ignored). We further depict $\overline{\xi}$ as a function of $\lambda$ for various system sizes. Notably, it becomes evident that the localization phase transition points for both $\Delta = 0.5$ and $\Delta = 1$ conform to the relational equation: $\lambda = 2\Delta$. Remarkably, these phase transition critical points remain invariant with the increasing system size. This large $\eta$ situation reminds us of the self-dual relationship of the AAH model. The difference in localization properties between $\Delta = 0$ and $\Delta > 0$ leads to the production of a new re-entrant phase transition.

# 4 The intermediate phase and mobility edge

In addition to the REPT phenomenon, one can also notice dramatic fluctuation of $\kappa$ in the red triangle region of Fig. 2, which is neither characteristic of pure phase nor traditional intermediate phase, but of brand new phases and new mobility edges in this region. The core difference between the new mobility edge and the traditional one lies in the critical phase. To discuss this type of region more comprehensively, next we will examine the $\Delta = 1$ and $\Delta = 1.2$ cases, which emerge with all types of new mobility edges. To distinguish from the traditional intermediate state (where the extended and the localized states coexist), we categorized all the possible intermediate states in the system formed by the critical state and other states into several types and named each one of them. Their composition and terminology are summarized in Table 1.

| Int. Phases | Components |
|---|---|
| Int. I | Extended + Localized |
| Int. II | Extended + Critical |
| Int. III | Localized + Critical |
| Int. IV | Extended + Localized + Critical |

Table 1: The intermediate (Int.) phases.

## 4.1 The case of $\Delta = 1$

$\Delta = 1$ is the critical case for the superconducting paired AAH model, which has only $\lambda = 4$ one transition point between the critical and localized phases ($\eta = 0$ corresponds to the blue dashed line in Fig. 1). We show the $\kappa$ and average IPR $\overline{\xi}$ in the $\lambda - \eta$ plane in Fig. 6(a)(b), respectively. One can see a boundary in the figure, i.e., the white dashed line of $\lambda = \eta$. This boundary divides the region where $\kappa$ has obvious fluctuations and another region where $\kappa$ is relatively stable, which means that there will be different intermediate phases (with or without

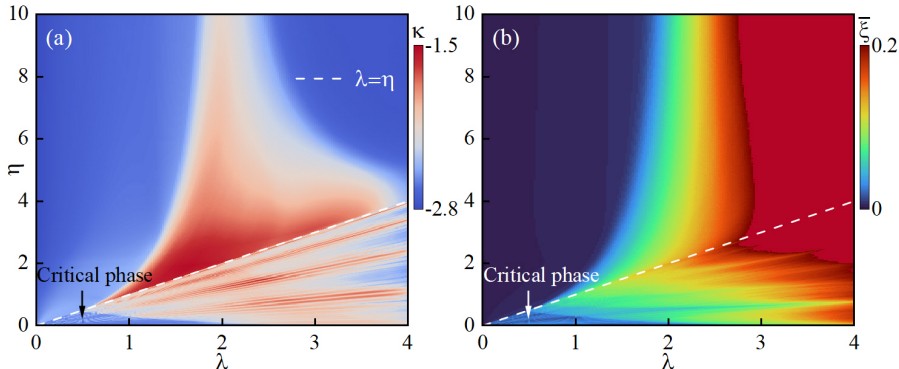

Figure 6: (color online). (a) $\kappa$ and (b) $\overline{\xi}$ show the $\lambda - \eta$ phase diagram for $\Delta = 1$. The white dashed line denotes $\lambda = \eta$. For all plots, the system size $L = 610$.

participation of the critical phase) on both sides of this critical boundary. Detailed evidence will be provided in the following part. Note that, in addition to the rich intermediate phases in the system, the results of $\kappa$ and $\overline{\xi}$ jointly reveal that when the staggered potential strength $\eta$ and the quasiperiodic strength $\lambda$ are relatively weak (The area labelled in the lower left of the figure), there still exists a pure critical phase.

To prove the above conclusion, we further discuss the fractal dimensions $\Gamma$ for different $\lambda$. Firstly, in Fig. 7(a), we show how $\Gamma$ corresponding to all eigenstates in the system changes with $\eta$ when $\lambda = 0.8$ is fixed. It is not difficult to notice that in the region with small $\eta$, $\Gamma$ behaves neither as an extended state ($\Gamma \to 1$) nor as a localized state ($\Gamma \to 0$), but somewhere in between, which is evidence for the existence of a critical phase. As $\eta$ increases, some of the eigenstates are localized. Interestingly, as $\eta > \lambda$, the critical state is instantaneously extended and thus enters the conventional intermediate phase. Specifically, the system undergoes the following phase transitions: critical phase $\to$ Int. III $\to$ Int. I $\to$ extended phase. Further, we selected $\eta = 0.2$, $0.5$ and $2$ in Fig. 7(b1)(c1) to discuss the fractal dimension $\Gamma$ at different system sizes. Since regions of $n/N \in [0, 0.5]$ and $n/N \in [0.5, 1]$ are symmetric, we exhibit only the results of region $n/N \in [0, 0.5]$. For $\eta = 2$ ($\eta = 0.2$), the $\Gamma$ of all eigenstates increases (invariably) with the system size, which is a good proof of the extended property (critical property) of the system. While for $\eta = 0.5$, the $\Gamma$ well characterizes the localized states (decreasing with the increasing system size) and the critical state as the system size increases, indicating that the system is in the Int. III phase.

Since the number of eigenstates will increase with the growing system size, to grasp the more accurate scaling behavior of the system, we define the average fractal dimension in region $R$ as

$$\overline{\Gamma}_R = \frac{1}{L_R} \sum_{n \in R} \Gamma_n, \tag{12}$$

where $L_R$ is the eigenstates' number of region $R$, and $R = loc, cri, ext$ correspond to the extended, localized, and critical regions, respectively. For extended (localized) states, the average fractal dimension $\overline{\Gamma}_{ext}$ ($\overline{\Gamma}_{loc}$) tends to be $1$ ($0$) as the system size $L_R$ increases, while $\overline{\Gamma}_{cri}$ corresponding to the critical state falls between $0$ and $1$ under the scaling limit. As shown in Fig. 7(b2)(c2), in the thermodynamic limit, $overline\Gamma_R$ with $\eta = 2$ is able to reach 1, while $\eta = 0.2$ lies between 0 and 1, indicating that it is in the extended and critical phases, respectively. For $\eta = 0.5$, the $\overline{\Gamma}_R$ of the critical and localized regions in the thermodynamic limit are around 0.6 and 0, respectively, indicating that the system is in the Int. III phase.

Secondly, we show the case of $\lambda = 3$ in Fig. 7(d)-(f). The results show that in the region $\eta < \lambda$ ($\eta > \lambda$) the system is in Int. III phase (Int. I phase). As $\lambda$ increases, the system

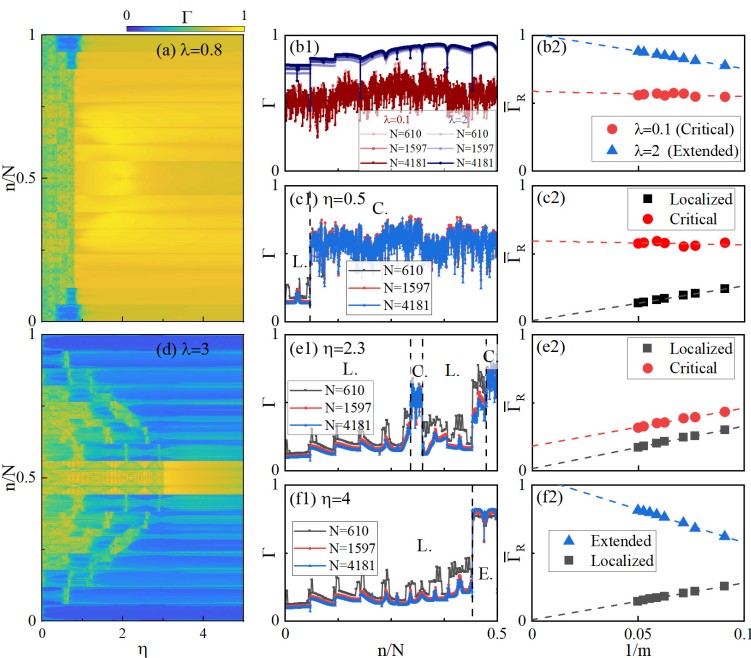

Figure 7: (color online). (a) Fractal dimension $\Gamma$ of all eigenstates as a function of $\eta$ for (a) $\lambda = 0.8$ and (d) $\lambda = 3$ with $N = 610$. (b1), (c1), (e1) and (f1) exhibit fractal dimensions $\Gamma$ as a function of $n/N$ for different system, where L., C., and E. are abbreviations of the Localized region, Critical region, and Extended region, respectively. (b2), (c2), (e2) and (f2) show average fractal dimensions $\overline{\Gamma}_R$ as a function of $1/m$ for different regions. Throughout, we set $\Delta = 1$.

undergoes a phase transition from Int. III phase to Int I phase at the critical point $\eta = \lambda$. Due to the multifractal property of the wavefunction, the different critical states have large numerical fluctuations. Therefore, one can see that the $\kappa$ of new type intermediate phases will be quite different from that of the conventional intermediate phase. But exactly which type of intermediate phase it is going to be indeed needs further discussion.

## 4.2 The case of $\Delta = 1.2$

Then we consider the more general case of $\Delta = 1.2$ with the relevant phase diagram shown in Fig. 8(a). When $\eta = 0$ (the case of red dashed line in Fig. 1), the system can be reduced to the standard $p$-wave superconducting paired AAH model. In this condition, the extended-critical phase transition and the critical-localized phase transition of the system occur at $\lambda = 0.4$ and $\lambda = 4.4$, respectively. The introduction of staggered potential will result in the emergent intermediate phases in the region of $\lambda \in [0.4, 4.4]$. Fig. 8(a) shows that $\kappa$ fluctuates in the region with weak staggered potential, and the fluctuation will become more significant as $\lambda$ increases, which suggests the occurrence of intermediate phases where different states coexist. To ascertain what exactly these intermediate phases are, we calculate the corresponding fractal dimensions with respect to different staggered potential. As shown in Fig. 8(b) for $\lambda = 2$, with the increase of $\eta$, special phase transitions occur. To be more specific, the phase transitions occur gradually as follows: Int. III phase → Int. IV phase → Int. III phase → Int. IV phase → Int. II phase → extended phase.

Fig. 8(c)-(h) show the results of the eigenstate scaling analysis. Here, scaling behaviors of fractal dimensions are discussed for $\eta = 0.5$(c)(d), $0.7$(e)(f) and $5$(g)(h), respectively. Specifically, when $\eta = 0.5$, the fractal dimension will be alternately of the localized phase

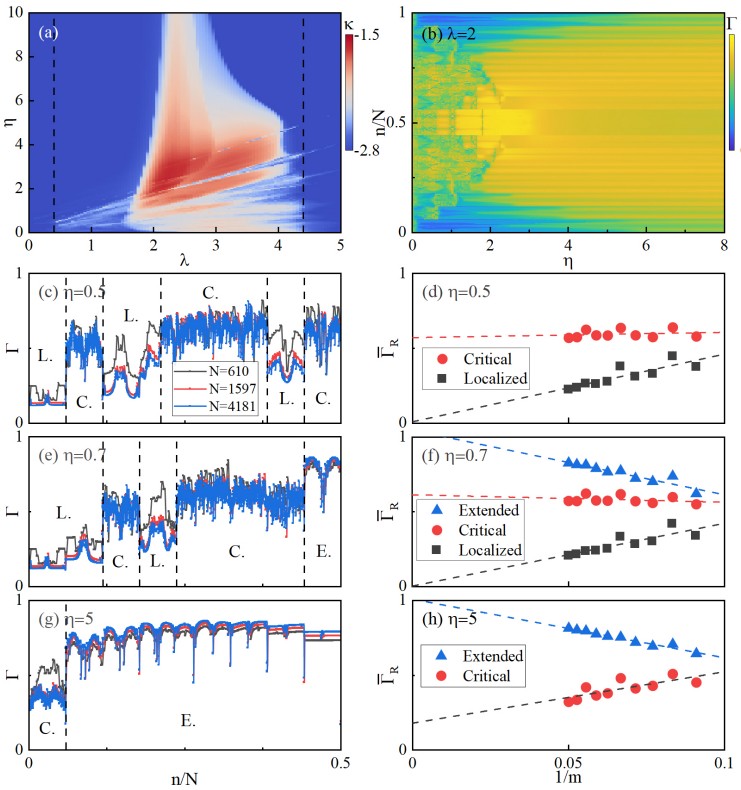

Figure 8: (color online). (a) The $\kappa$ phase diagram in $\lambda - \eta$ plane with $\Delta = 0.5$. The black dashed lines correspond to slices with $\lambda = 0.4$ and $\eta = 4.4$, respectively. The two black dashed lines mark the critical points of phase transition from the extended to critical and from critical to localized phases of standard $p$-wave paired AAH model ($\eta = 0$), respectively. (b) Fractal dimension $\Gamma$ of all eigenstates as a function of $\eta$ for $\lambda = 2$ with $N = 610$. (c), (e) and (f) exhibit fractal dimensions $\Gamma$ as a function of $n/N$ for different system, where L., C., and E. are abbreviations of the Localized region, Critical region, and Extended region, respectively. (d), (f) and (h) show average fractal dimensions $\overline{\Gamma}_R$ as a function of $1/m$ for different regions. Throughout, we set $\Delta = 1.2$.

and the critical phase as $n$ increases, which is the evidence of Int. III phase. When $\eta = 0.7$, the fractal dimension indicates the coexistence of the localized, critical and extended states in the system, which proves the existence of Int. IV phase. Finally, when $\eta = 5$, the fractal dimension shows that the critical phase and the extended phase coexist in the system, which is the evidence of Int. II phase.

Further, the scaling analysis of the fractal dimension $\overline{\Gamma}_R$ for different regions is shown in Fig. 8(d), (f) and (h), and the average fractal dimension in different regions again confirms the existence of three intermediate phases, i.e, Int. III, Int. IV and Int. II phases.

# 5 Conclusion

In summary, we investigate the $p$-wave paired quasiperiodic model with staggered on-site potential. On the one hand, we report for the first time the reentry to the extended phase with increasing quasiperiodic intensity, i.e., the REPT phenomenon. Furthermore, we prove that multiple REPT phenomena can emerge in the system with varying staggered potential

strength. On the other hand, different from the traditional intermediate phase (composed only of the extended and localized states), we find that there are novel intermediate phases in the system, which contain the critical states. Through the fractal dimension and scaling behavior analysis, we prove that there are three types of intermediate phases, namely, the extended state + the critical state; the localized state + the critical state; and the extended state + the localized state + the critical state, respectively. Since quasiperiodic models have already been successfully implemented in various tabletop experiments [25–41], the REPT phenomena and intermediate phase predicted in this paper are expected to be observed in the near future.

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
