# Peer review of "Emergent Recurrent Extension Phase Transition in a Quasiperiodic Chain"

_SciPost Physics_

## Round 1 · Referee Report · Xianlong Gao (Referee 1) · 2025-9-8

Strengths

The manuscript studies a p-wave superconducting AAH chain with an additional staggered on-site potential. Using inverse and normalized participation ratios (IPR/NPR), fractal dimensions, and multifractal scaling indices, the authors report: 1 a “recurrent extension phase transition” (REPT): re-entrance from an intermediate phase back to a fully extended phase upon increasing either the quasiperiodic strength λ (Type-I) or the staggered strength η (Type-II), at relatively weak potentials; 2 the emergence of three intermediate phases comprising critical states in addition to extended and/or localized states (Int. II–IV), in contrast to the “traditional” intermediate phase (Int. I) containing only extended and localized states; 3 in the large-η limit, a sharp extended–localized boundary that appears to approach λ = 2Δ for Δ > 0 (while for Δ = 0 they argue λc ≃ 1/η for large η).

Weaknesses

To meet the standards of SciPost Physics, the manuscript requires substantial strengthening in terms of conceptual clarity, evidence for the main claims (especially REPT) and analytical support.

Report

The topic is timely and relevant to current interest in multifractality, mobility edges, and re-entrant phenomena in quasiperiodic systems. The numerical machinery is standard and appropriate, and the observation that critical states participate in the energy-resolved phase composition in this model class is plausible. The paper can be accepted.

Requested changes

  1. The paper positions REPT as conceptually distinct from previously reported “reentrant localization transitions” (repeated localization as parameters vary) and highlights intermediate phases featuring critical states. However, coexistence involving critical states and multi-mobility-edge structures have been reported in related contexts (e.g., spin–orbit, off-diagonal quasiperiodicity, and non-Hermitian settings). Please refine the novelty statement to clarify precisely what is new: is it the “return to fully extended” in a real, Hermitian, BdG AAH + staggered setting; the classification Int. II–IV in this model; and/or the large-η asymptote? Provide a more explicit comparison to Refs. [47-48, 55, 84–92,106-113] and explain how your phenomenology differs. A deeper comparative analysis would strengthen the novelty claim. 2 The numerical methods (e.g., IPR, NPR, fractal dimension, scaling analysis) are standard and appropriately applied. The phase diagrams (Figs. 2,6,8) and scaling behaviors (Figs. 3,4,7,8) provide solid evidence for REPT and new mobility edges. The thermodynamic limit extrapolations (using Fibonacci numbers) are handled well. However, some claims lack sufficient justification—e.g., the "Type-I" and "Type-II" REPT distinctions feel ad hoc without analytical backing. An analytical derivation (perhaps via duality transformations, as in Refs. [44,51,58-60]) could elevate the work. Additionally, the limit of large η (Sec. 3.4) is interesting but underexplored; why does REPT disappear, and is there a perturbative explanation? 3 The model (Eq. 1) is clearly defined, and parameters (e.g., N=610, α=F_{m-1}/F_m, θ=0) are specified. The Bogoliubov-de Gennes transformation is correctly implemented. Calculations are done with PBC. In p-wave chains, OBCs reveal Majorana edge physics and a topological phase structure that may interplay with localization. At minimum: Comment on whether REPT persists under OBC and whether Majorana zero modes survive across the REPT windows. If feasible, provide a brief OBC check (even small N) showing similar alternation in bulk-state indicators, and state whether edge modes complicate IPR/NPR diagnostics near zero energy. 4 I suggest to try the attempt a self-dual approximation or Avila's global theory (Ref. [73]) to derive REPT/phase boundaries and expand on new intermediate phases (Fig.1 is a repeated phase diagram and should be cited correctly): for example, provide wavefunction visualizations or energy spectra to illustrate Int. II-IV phases. 5 Cite and compare with recent works (e.g., arXiv preprints in Refs. [57,63,110,112]) more critically. 6 Please define REPT operationally (e.g., in terms of thermodynamic-limit indicators) and specify an objective procedure for demarcating the intervals where the system is intermediate vs extended. Currently, κ = log10(ξ×ζ) is used qualitatively with a heuristic “pure phase ≈−3” threshold. Provide size scaling of κ distributions and a finite-size extrapolation of the REPT boundaries to show the phenomenon is not a finite-size effect. 7 The manuscript states that θ “has no qualitative impact,” and sets θ = 0 throughout. For mobility-edge and multifractal properties, θ can materially affect finite-size and sample-to-sample fluctuations, especially in weak-potential regimes. Please add θ-averaged diagnostics (e.g., averaging κ, ξ, ζ, and Γ over a modest set of θ values) and show that REPT persists statistically. Alternatively, provide a figure showing the θ dependence at representative parameters spanning the REPT regions. 8 The classification into Int. II–IV is interesting, but the presentation is qualitative. To justify claims of “novel mobility edges,” provide energy-resolved phase maps: Plot \gamma_n (or \beta_min) vs energy En for multiple system sizes and extract mobility-edge curves Ec(\lambda, \eta) where the phase switches (e.g., \gamma_n crossing thresholds extrapolated to N → ∞). Provide histograms or cumulative distributions of \gamma_n over energy to visualize the coexisting fractions of each phase and quantify them (fractions vs \lambda, \eta). 9 For the large-\eta limit: need for an analytical argument and careful verification. The large-\eta result is a highlight but remains heuristic. The claim that for \delta = 0 the transition occurs at \lambda c ≈ 1/\eta at large \eta needs either a reference with matching conventions or a derivation. Likewise, for \delta > 0 the assertion that the boundary approaches \lambda = 2\delta (apparently independent of \eta and J once \eta is “large enough”) deserves an analytical rationale. 10 Provide an effective model in the \eta → \infty limit (e.g., via a Schrieffer–Wolff transformation to a dimerized sublattice-effective model). Derive the effective hopping/pairing amplitudes and their \lambda dependence. From that, obtain an analytical criterion for the localization boundary, and compare to numerics. 11 Clarify under what scale separation (“how large is large?”) the asymptotic relation \lambda_c = 2\delta holds numerically, and provide error bars/finite-size trends. 12 The manuscript reports the phenomenon but offers little physical intuition for why extended states reappear as \lambda or \eta further increase within the weak-potential regime. So, please provide a qualitative mechanism (e.g., competition between the alternating on-site staggering and p-wave pairing that reshuffles the effective band structure and renormalizes the quasiperiodic modulation seen by Bogoliubov quasiparticles). If possible, show a simple k-space toy model or an approximate self-duality/double-resonance argument that can explain alternating windows of enhanced delocalization. 13 Discuss whether REPT is tied to commensuration effects between the staggered period 2 and the incommensurate modulation, possibly leading to parametric windows where backscattering is suppressed for a subset of BdG bands. 14 Typos are "participation ration", “sacling index”, "obtian"; page 3, first line: \sum; Inconsistent use of "quasiperiodic" vs. "quasi-periodic"; in Fig. 8 caption (a) with Δ = 0.5, contradicting with “throughout, we set Δ = 1.2.

Recommendation

Publish (meets expectations and criteria for this Journal)

---

## Editorial Decision

in_refereeing